



# Grand Challenges in Social Aspects of Wind Energy Development

Lena Kitzing[1], David Rudolph[1], Sophie Nyborg[1], Helena Solman[2], Tom Cronin[1], Gundula Hübner[3], Elizabeth Gill[4], Katherine Dykes[1], Suzanne Tegen[5], Julia K. Kirkegaard[1]

[1]Department of Wind and Energy Systems, Technical University of Denmark, Risø Campus, Roskilde, 4000, Denmark
[2]Department of Social Sciences, Wageningen University and Research, Wageningen, 6705, The Netherlands
[3]Institute of Psychology Martin-Luther-University Halle-Wittenberg, MSH Medical School Hamburg University of Applied Sciences and Medical University, Hamburg, 20457, Germany
[4]Accelerated Deployment and Decision Support Center, National Renewable Energy Laboratory, Flatirons Campus, Arvada, 10  Colorado, 80007, USA
[5]Center for the New Energy Economy, Colorado State University, Powerhouse Energy Campus, Fort Collins, Colorado, 80523, USA

*Correspondence to*: Lena Kitzing (lkit@dtu.dk)


**Abstract.** Social aspects are gaining traction in wind energy research. Increasing local opposition to wind energy projects is just one symptom of deeper-rooted challenges in the further expansion of the technology. A recent publication by Kirkegaard et al. (2023a) lays out the grand challenges related to the complex interactions between society and wind energy technology and outlines a research agenda for wind energy research from a socio-technical perspective. This article discusses these 20  challenges in the context of a more technologically focused research audience. We begin by describing the role of social sciences in wind energy research, arguing for the diverse set of insights, research topics, and value that they can add, going beyond outdated concepts of social acceptance (such as NIMBY), and providing solutions for public engagement and planning processes, just ownership structures and value-based design. We discuss social grand challenges in five areas: (1) Project planning & spatial relations, (2) Wind turbine design & scalability, (3) Grid integration, roles & responsibilities, (4) General 25  public perception of the technology, (5) Energy policy to support system transformation. We conclude by reflecting how social sciences and technical sciences can be better integrated to jointly advance wind energy research into a new interdisciplinary era that is able to provide holistic solutions for a transition to carbon-neutrality.

## 1 The challenge

Energy systems around the world are in an exhilarating development: projections show that there is a doubling or even tripling 30  of global electricity production required by 2050, to tackle the two-fold challenge of creating a zero-carbon energy system and ensuring adequate energy access for everyone (Kitzing et al., 2022). The zero-carbon pathways developed by international experts foresee wind energy to contribute more than half of the electricity production in Europe and around a third of electricity production in the world by 2050 (EC, 2018; IEA, 2021). In a few countries, e. g. Denmark and Spain, wind energy has already taken its place as a major contributor. In many other countries, the deployment of wind technology is still in its infancy. The



foreseen eleven-fold increase in installed capacity (EC, 2018) implies that wind energy will reach scales and locations that by far exceed anything we have seen until now.

The expanding footprint of wind energy has inevitable social and environmental implications. Wind energy infrastructures will likely become increasingly visible, change our landscapes and affect the way in which people use and relate to these new

energy landscapes (Stremke et al. 2022). The technology will interact more with human communities, in rural and urban areas, inland as well as in coastal areas. Some previous strategies like moving wind turbines away from where people live or where no stakeholders are impacted, seem no longer feasible or plausible. There is an increasing recognition that, with the changed role that wind energy will play in the future energy system, further expansion targets cannot be achieved by technological solutions alone: "Success to date has been achieved by engineering […], with solutions often driven by experience as much as

by fundamental understanding. Unfortunately, this approach can no longer provide the innovation demanded to support the systems of the future" (Veers et al., 2022, p. 2492). Instead, we will have to explore new ways of reconciling wind turbines with the livelihood of people by creating mutual benefits, by establishing a productive relationship between wind farms and people, by exploiting the constructive value of conflict and controversy (Cuppen, 2018), and by granting people more power for co-shaping different elements in the process of developing and deploying wind farms. This requires a fundamental

rethinking of the relationship between humans and wind turbines, of the design and deployment of wind energy in our communities.

## 2 The role of social sciences in wind energy research

Social sciences are an important means to understand and tackle socio-technical challenges in wind energy. Social sciences study societies, individuals and their behaviours, as well as the relations among individuals (or groups of individuals) within

those societies. Energy-related social sciences seek to better understand the interactions between humans and energy production, supply and use, in particular of the transition to a low-carbon energy system, by studying social phenomena (such as norms, values and valuations, concerns, perceptions, institutions, practices, etc.) that shape human-energy-interactions, and related fundamental issues of equity, fairness, ethics, or attribution (Dufour et al., 2019) as well as perceived annoyance (Hübner et al., 2019). Social Sciences and Humanities (SSH) are often contrasted with STEM (Science, Technology,

Engineering, Mathematics). SSH comprise a broad range of disciplines (more than twenty), from Anthropology over Economics, Education, Geography, History, Law, Political Sciences, Psychology, Sociology, to Science & Technology Studies (STS). The most represented disciplines in a recent study of social science research in renewable energy were: Sociology, Political Sciences, Human Geography and Economics (Krupnik et al., 2022).

Social science research on wind energy often seeks to understand the human aspects of technology development and implementation, assuming that the relationship between humans and technology is inherently political, complex and



intertwined in a web of relations among different societal actors, technologies and landscapes, as well as dynamic and context specific. Additionally, social science collaborates with technical science to analyse the impact of wind turbine on wellbeing and seeks to derive mitigation measures (Pohl et al., 2012; Müller et al., 2023). Social science research often explores and
learns from past, present or future wind energy developments, documenting single or multiple cases and shining light on their similarities and differences in various cultural and political contexts (Ellis & Ferraro 2017; Ellis & Määttä 2023). Growing local opposition to wind energy has also sparked a whole line of research about societal acceptance of wind energy (e.g. Wüstenhagen et al. 2009; Batel 2020; Hübner et al., 2023).

Thanks to a variety of methods and conceptual approaches, SSH studies can offer multiple relevant perspectives on wind energy (see also Sovacool, 2014). Methods applied include a variety of approaches, tools, and techniques for collecting and analysing qualitative and quantitative data. They include (amongst others) field surveys, ethnographic research, case studies, experiments, statistical analysis, simulations, and conceptual considerations. For example, different SSH studies have collected a broad range of evidence about why wind turbine noise is a matter of societal concern and how different individuals perceive
wind turbine sounds (Müller et al., 2023; Solman et al., 2023), why wind turbine noise is a topic of controversy (e.g. in relation to impacts on human health (Taylor and Klenk, 2019), and how communities oppose wind energy because of noise annoyance (Fast et al. 2016). At the same time, SSH research also often attempts to research possible solutions to such problems – e.g. discussing the implications of different planning choices (Lienhoop, 2018), developing briefings for technical mitigation measures (Pohl et al., 2012, 2021), and providing policy advice (Kemp and Rotmans, 2009).


Some may fear that an increased involvement of SSH could make the development of wind energy more complicated or difficult. The fact is, however, that social sciences do not increase complexity – they uncover and describe complex realities; they clarify, categorise and inform about issues. Underlying the social science approach on the topic of wind energy is a fundamental interest in understanding how and why societal actors respond in a certain way to wind energy and how this
knowledge can be used to better govern the energy transition. Insights from social science studies can be used to improve design and implementation of wind energy is a way that is both technically feasible and socially desirable. For example, SSH can contribute solutions for smoother and faster deployment processes that lead to fewer contestations. Numerous studies provide useful insights about how to foster public engagement in planning processes (Fast, 2017), how to encourage just ownership structures (Jacobsen, 2008) and how to enable technology design that reflects public concerns and values (Solman
et al., 2022). In this way, social science literature provides information about socially desirable factors for customisable solutions, scale, sizing, siting and operational control.

Nonetheless, social issues are still sometimes seen as a barrier that arises during the planning and installation of a project (Sovacool et al., 2015, Solman 2023). Solutions to this barrier are too often still sought to be found in immediate, short-term
'fixes' that silence protesters and make local communities accept the project. For example, residents' concerns about the visual



impact of wind farms are tackled by providing information about the design of future wind farms, rather than involving communities in their co-design (Oosterlaken, 2015). Such a mindset, however, fails to acknowledge the wicked nature of the problem and regrettably sometimes amplifies future social issues by antagonising residents of wind farm areas.

Truly socially sustainable solutions will need to venture beyond decontextualised, instrumental techno-centric assessments of acceptance and outdated not-in-my-backyard (NIMBY) concepts (Devine-Wright 2005; Batel and Rudolph, 2021). This ultimately means leaving the instrumental understanding of social acceptance behind that has dominated our thinking for too long (see also Kirkegaard et al., 2023a). Social sciences can offer a much more diverse set of insights into the challenges of wind energy development than providing a remedy for issues of local opposition. This requires independent (but interrelated)

social science research, i.e. we need social science scholars focused on wind energy technology who set their own agenda and ask research questions founded in their discipline's perspectives around how, where and by whom should wind energy technologies (not) be designed, planned and developed (Solman et al., 2021; Sovacool 2014). Moreover, at the fundamental level, the goal and need for social science research is to understand and define the issues in wind energy and how they are inevitably political (Krupnik et al., 2022).

**3 The grand social challenges**

A number of different challenges have been identified that require social sciences research along the full life cycle of wind energy, including design, planning, development, operation, and end-of-life. In a recent article, Kirkegaard et al. (2023a) describe several social grand challenges of wind energy in detail and demonstrate the advantages of integrating social science perspectives along the different life phases of wind energy development. We will in the following discuss and reflect on some

grand challenges and the role of social sciences from a different perspective within the context of wind energy research.

**3.1 Project planning and development & spatial relations**

The social (or 'human') dimension introduces different locational and territorial challenges that can constrain the siting and sizing of wind turbines and projects. These may contradict physical wind resource considerations which is why social science literature can offer insights about where, how and by whom wind energy projects should be developed. Local resistance to

wind farm developments is more and more common. This phenomenon still tends to be simplified and labelled as selfish NIMBY-behaviour. However, Kirkegaard et al. (2023a) clarify that local resistance and annoyance has now been rethought as representing responses to disruptions related to place identity, place attachment, spatial-visual perceptions, as well as reactions towards issues of spatial justice (see e.g. Devine-Wright & Howes, 2010; Firestone et al., 2018; Kim & Chung, 2019; Batel & Devine-Wright, 2021). Wind energy can interfere with human well-being through physical 'emissions' (sound, light, shadow

flicker, etc.) (Hübner et al., 2019; Pohl et al., 2018), as well as with relations and emotions that humans have towards land and sea (Russell & Firestone, 2021). Feelings of injustice may arise, e.g. resulting from unjust practices in land acquisition (land





grabbing) (Kirkegaard et al., 2022, Jacquet, 2015), dispossession of local land, (perceived) uneven distribution of wind energy 'burden' across jurisdictions and of value creation from it (Bosch & Schmidt 2020). General changes in the landscape due to the deployment of wind energy include not only visual change, but also indirect effects such as de-population, replacement
and/or incoming of different types of industries, and land use changes (Rudolph & Kirkegaard, 2019).

Social scientists have started to advocate for seeing local communities, citizens and lay persons as 'experts' in their own right. These people can share their local knowledge about the impact that the technology causes, what the community values and despises, and what effects and contestations may emerge – and, if allowed, they can become important allies and develop
creative solutions. Social sciences research can help to better understand the human experience with wind energy in people's everyday lives and develop solutions to improve it. Much more data on impacts and human responses will need to be collected, also across projects, i.e. going beyond project-by-project case studies. This should also include indigenous populations and the Global South. An improved understanding of cross-cutting issues for impacts and responses (correlations) and methods on how to derive general learnings from increasing numbers of single case studies and findings for specific contexts must be
developed. The improved understanding will facilitate the development of tools for plant management and control strategies adapted to the needs of neighbours. Distributional effects of wind energy (benefits and burdens) need investigation, and solutions for improved fairness need to be developed, e.g. including benefit sharing schemes and their implementation procedures.

### 3.2 Wind turbine design & scalability

There is a growing anxiety by local communities about visibility, noise impact (Müller et al., 2023) and dominance of wind turbines (Batel and Devine-Wright, 2021). Social science literature provides a nuanced understanding for how these problems should be understood and tackled. Turbines have outgrown the human scale, creating a perceived mismatch between energy infrastructure and the landscape. Voices are emerging that challenge the prevailing innovation paradigm of 'bigger is better'. Kirkegaard et al. (2023a) discuss how increasing the size of wind turbines has been part of the engineering ethos in the wind
energy sector since its beginnings in the 1970s, motivated by an economic logic of increasing energy yield, cost reduction, and profit-increase. The authors point out that such research focus seems today somewhat paradoxical or outdated as onshore wind energy has become cost-competitive against fossil fuels (Kirkegaard et al, 2023a; Kirkegaard et al. 2021). With large scale turbine designs dominating the market, there is currently little space for developing designs in a wider range of directions that can address different ways of value creation, for instance towards urban wind or other small-scale designs, that may, in fact,
be more adapt in keeping with the values of the wider population and create significant additional deployment potentials and enhance the societal value of wind energy (see Lunevich & Kloppenburg, 2023).

Turbines currently available on the market are often not affordable for non-commercial actors, marginalising small investors who would like to participate in the energy transition (Kirkegaard et al. 2021; Kirkegaard et al. 2023b). Difficulties in engaging





societal actors with utility-scale wind energy projects are creating new issues of justice. Social science research can help to open up the discussion of 'what matters' in developing technology further and acknowledging the potentially conflicting interests of various actors and their diverse agency in the process (Kirkegaard et al. 2023a). Bringing in non-technical experts and citizen representatives into the design process of future technology in a co-creation process may enhance the ability to deploy future turbine designs (Solman, 2022). Here, solutions must still be developed as to how to meaningfully engage societal

actors in the design of wind turbines, and in particular also how to involve societal actors in the ongoing wind energy digitalisation process, so that new tools reflect their interests.

### 3.3 Grid integration, roles & responsibilities

The adequate integration of wind energy into the electricity grid is a precondition for the further expansion of wind energy. In this context, 'smart grids' are a key topic (e.g. Nyborg and Røpke, 2011; Strengers, 2013) within SSH literature. The

engineering vision of the smart grid often entails specific images of the role users should play in the future grid (Silvast et al., 2018), i.e. become flexible in their consumption to balance intermittent energy production, motivated by an economic rationality. However, these images, which are embedded in smart home technology designs, do not always match the reality of the everyday life of consumers, and may even be counter-productive to a sustainable development (Nyborg and Røpke, 2011, Nyborg & Røpke, 2013). Several social science studies point to how users feel a lack of control and agency around

energy use (Adams et al, 2021), and how users may make 'hacks' to the systems so that they fit to their reality (Hyysalo et al., 2013, Nyborg 2015). Also, transparency around (big) data ownership, data sharing and privacy issues pose SSH concerns to be addressed (Hand, 2018, Kloppeburg et al., 2022). Opportunities to develop and participate in (digital) energy communities are mostly an opportunity for those with the resources and ability to invest (Kirkegaard et al., 2023a, 2023b).

Social science research helps exploring the major reconfigurations in roles and responsibilities as well as the emergence of new business models that new smart system integration designs and energy community policies promote, related to notions such as prosumers and aggregators (Kloppenburg and Boekelo, 2019). The concept of 'community' needs further research attention in relation to new roles for citizens, e.g. providing balancing services and flexibility via micro-generation. While flexibility and prosumption dynamics have been explored at the individual household level (e.g. Hansen and Hauge, 2017,

Christensen et al 2017, Nyborg 2015), questions as to how communities organise and negotiate this role collectively are important future research topics (e.g. Astola et al, 2022). Likewise, questions of energy community mobilisation, and how to balance community and system needs, merit further attention. Finally, studies are needed that explore how new digital energy communities and smart technologies bring forward public and political issues, enabling 'material participation' (Marres 2015, Ryghaug et al., 2018), and new forms of engagement with the electrical grid (Elkjær et al, 2021).





### 3.4 General public perception of the technology

On a broader societal level, SSH literature shows that we experience a change in the interactions between people and wind energy technology. Formerly, it was seen as a grass-root technology and environmental success driven by green enthusiasts against fossil incumbents (Kirkegaard et al., 2023b). In many countries, it was mainly farmers and private citizens who drove the early wave of installations of small wind turbines. However, now contestations are arising on grounds of a commodification and marketisation of wind energy (Kirkegaard et al., 2021). Addressing people-technology relations at a general level could include making the technology development a 'public issue', including co-creation (Solman et al., 2021; Batel & Rudolph, 2021) and material participation (Lenoir-Improta & Di Masso, 2021). Social sciences can help wind energy experts and societal actors to come together to discuss and co-design wind energy systems of the future. Social sciences research may also have to look 'inwards' into the wind energy sector to understand the perception and reactions of the sector toward their changed relations with the public. It is not an easy task to master the transition from 'welcome saviour' who was perceived as green ally in preventing climate change to (in some places) 'suspicious intruder' who is expected to make an effort to win people's acceptance, prove sustainability, minimise environmental impacts and to create additional societal value in multiple dimensions (Kirkegaard et al., 2023a).

Social science research must map and understand the attitude and valuations toward and opinions regarding wind energy technology itself as well as of its use of materials and space. This also includes global supply changes, and e.g. the extraction of raw materials in the Global South, related justice and inequality questions and lingering relations to colonialism. The risk of locking-in technology in cultural and societal misalignment must be understood and assessed better. Social sciences can help developing co-creation methods to ensure that future innovations drive the technology towards improved cultural and societal alignment, so that e.g. the scale and look of wind turbines and wind farms are better embedded into society's specific needs wherever they are to be placed.

### 3.5 Energy policy to support system transformation

Wind energy development has always been strongly policy-driven (Kitzing et al., 2022). Whereas wind energy policy could previously focus on nurturing the technology in its niche by providing support, social science studies offer insights into how the challenges are now much more complex and diverse, so that wind energy policy must evolve alongside the technology. Renewable energy technological development is uncertain, dynamic, systemic and cumulative (Grübler, 1998). Policy and regulation are also shaped by uncertainties and barriers arising from market conditions, existing policies, the legal system, policy-making tradition and culture. Recent declines in the cost of wind energy diminish the need for support payments and emphasise a new focus on redesigning markets and employing new policy areas. Value creation across technologies will become more important, in addition to well-designed permitting and siting procedures, as well as speedy grid connection. Social science research helps to understand the energy transition as a comprehensive long-term transformation process that





extends well beyond the typical time horizon of political processes. At the same time, the existing energy system exerts a strong momentum for its own continuation (Hughes, 1987), locking-in existing technologies and policies and posing barriers for new technologies and practices (Unruh, 2000). There is currently little understanding of how to integrate systemic

requirements for wind energy policy with multiple policy objectives that also play an increasing role in political decision making, including domestic industry development, job creation, workforce development, strengthening of technology export capabilities, strengthening of supply chains, and increasing energy security.

Social science studies are needed to provide policy support that is nuanced and focused on serving a broader array of social

objectives. This relates to creating enabling environments (by e.g., creating long-term stability and transparent policy goals, ensuring adequate electricity networks) and adapting frameworks (by e.g., adapting market designs, driving electrification and digitalisation) (Kitzing et al., 2021). Social science research can create a better understanding of social costs, benefits and effectiveness of policies, laws and regulations. It supports the systematic development of policy frameworks that are flexible, adaptive and reflexive (Voß et al., 2009) and that promote 'win-win' ideas for all stakeholders (developers, manufacturers,

local industry, local residents, citizens, etc.) (Kitzing et al., 2021). Quantitative economic analysis as well as qualitative policy studies help to evaluate the impact of energy policies on individual actors and systems, informing which factors drive deployment across different contexts, and how policies help reducing risks and to enable stable returns for project investments. Policy assessment and evaluation, empirical research and case studies inform how policies, laws and regulations are implemented and with how much success; they help to identify best practices and to find an adequate balance between the

need for clean energy and the impacts caused by specific projects.

## 4 Conclusions

Kirkegaard et al. (2023a) state that "wind energy technologies on their own do not matter much if they cannot be deployed *in* and *with* society" (p. 656). It is equally true that there is not much to deploy if there is no technically and economically viable technology available that keeps up with innovation needs. We are indeed in a truly mutual relationship. The changing realities

in which wind energy is deployed – also including its intricate relationship with the emergent hydrogen economy and Power-to-X technologies – requires a rethinking of current practices throughout the whole lifecycle of the technology. It requires a new generation of engineers and technical scientists who are skilled in and reflexive about social issues, and a new generation of social scientists who are trained in technical aspects and solution-oriented research (Kirkegaard et al., 2023a). The grand challenges of wind energy can only be solved together.


Wind energy science has in the past already demonstrated a remarkable capacity to integrate new strands of science whenever it was required. One of the most prominent examples is computational science and digitalisation, with exceptional applications of digital twin technology, e.g. in wind turbine design (see Solman et al. 2022). Today, numerous technical sciences are



working together every day to innovate and improve wind energy technology. A strong and continued effort has resulted in
wind energy science becoming a vital and acknowledged research focus for researchers from many STEM disciplines. In the
SSH area, this is not yet as prevalent. Universities still have a task to demonstrating clear career pathways for scholars from
different disciplines to specialise in technology application (enabling increasingly deep analysis over time) as well to dive into
the interdisciplinarity that is necessary for the creation of tailored solutions to real-life problems.

It is good news that the wind energy sector as well as the research community have started to acknowledge the task of
integrating technical and social aspects in research and development. Highly constructive dialogues have already taken place,
showing curiosity and emerging understanding with each other's disciplines and ways of thinking. The integration will not be
easy and surely some elements will be lost in translation. But with dedicated focus and the long-term goal in mind, a truly
interdisciplinary socio-technical perspective can be developed to tackle the grand challenges at hand. This can enable a new
era of interdisciplinary wind energy science across STEM and SSH disciplines that paves the way for a successful
transformation of our society to carbon-neutrality.

**Author contribution**

LK and KD conceptualised the idea and developed the format of the article; LK, DR, SN, HS, JKK, and TC wrote the original
draft, with contributions from GH; All reviewed and edited the drafts; JKK validated and oversaw the suitable reflection of
the main source article; LK administered the project.

**Competing interests**

At least one of the (co-)authors is a member of the editorial board of Wind Energy Science.

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
