# Peer review of "Grand Challenges in Social Aspects of Wind Energy Development"

_Wind Energy Science, 2023_

## Referee Comment (RC2)

**Review**

https://doi.org/10.5194/wes-2023-174

**Title: Grand Challenges in Social Aspects of Wind Energy Development**

**General comments**

The authors discuss several grand challenges in social aspects of wind energy development. Areas considered are (1) Project planning & spatial relations, (2) Wind turbine design & scalability, (3) Grid integration, roles & responsibilities, (4) General public perception of the technology and (5) Energy policy to support system transformation. The focus is mainly on the interconnection of technical and social aspects and how this can be taken more into account in the future since technology has dominated wind power development until now. Because of (global) risks like climate change, energy scarcity and growing international conflicts (e.g. about energy resources and land), the question of how to put a renewable energy system into practice is of central importance. As far as I can see it, addressing "grand challenges" with respect to this topic is very valuable and could add to significant progress in research as well as development.

I think two points have to be made clear right at the beginning of the paper. First of all, the normative perspective. Although it might seem obvious that we need renewable energy systems in future, there are still advocates of nuclear energy and perhaps even fossil fuels. On the one hand, this is due to business considerations. On the other hand, there are also scientifically discussed arguments like the potential of nuclear energy to combat climate change. Although this hypothesis has been empirically tested quite often (e.g. Bickerstaff et al. 2008, Spence et al. 2010, Pidgeon et al. 2008) and recent research showed that the public doesn't really believe in it (Sonnberger et al. 2021), decision makers in politics and the economy may think (and act) differently. Because of that, I recommend stating the normative position that we need renewable energy systems all over the world for a sustainable future at the beginning of the paper. The other point refers to the spatial scope of the paper. Is it meant to refer to the whole world, specific regions (e.g. western, industrialized countries) or single countries? There are of course great cultural, economic, political and administrative differences between countries and regions. Do the grand challenges really affect them in the same way? I think this point is worth of consideration and I would also accept an answer that states some limitations.

**Specific comments**

I also have some more specific questions and recommendations:

- The authors state on p.2 that "[…] we will have to explore new ways of reconciling wind turbines with the livelihood of people by creating mutual benefits […] by granting people more power for co-shaping different elements in the process of developing and deploying wind farms". I agree with that but how could this be done? Have you some ideas about that or is it simply stating more research need? The same applies for p.6, line 170-171.
- The authors are very enthusiastic on p.3: "Insights from social science studies can be used to improve design and implementation of wind energy is a way that is both technically feasible and socially desirable. For example, SSH can contribute solutions for smoother and faster deployment processes that lead to fewer contestations". I am not sure if it is that easy. For example, we know very well that fairness of the decision making process is of great importance for the implementation of wind energy projects (e.g. Hall et al. 2013, Hoen et al. 2019; Sonnberger und Ruddat 2017). But what is considered as "a fair process" isn't that clear in any instance. I recommend a more "realistic" or moderate formulation.
- The role of "[…] independent (but interrelated) social science research […]" is highlighted on p.4. I totally agree with that. We need it. But have you any ideas who will pay for it and why?
- Considering distributive fairness, the authors state that "distributional effects of wind energy (benefits and burdens) need investigation, and solutions for improved fairness need to be developed, e.g. including benefit sharing schemes and their implementation procedures" (p.5). This idea is not really new (see Ruddat 2022).
- What exactly does the concept of "urban wind" (p.5, line 159) mean? Are there "small-scale designs" thinkable and viable that would be accepted by the residents of a metropolitan area? And how much energy could be produced by such a design?
- "Social sciences can help wind energy experts and societal actors to come together to discuss and co-design wind energy systems of the future" (p.7, lines 203-204): Again, this view (or at least this formulation) is very optimistic. Of course, talking with each other helps a lot but not all conflicts can be solved that way. The possibility of a "consensus about dissent" (Renn 2004) has to be taken into account.
- What do you mean by the "[…] a truly interdisciplinary socio-technical perspective […]" on p.9, line 268-269? Maybe I didn't get it in the text. I worked in some interdisciplinary energy projects and all in all I had the feeling that we were working together (although I have to admit that there always was the risk of technology domination).

**References**

K. Bickerstaff, I. Lorenzoni, N.F. Pidgeon, W. Poortinga, P. Simmons, Reframing nuclear power in the UK energy debate: nuclear power, climate change mitigation and radioactive waste, *Public Understand. Sci.* 17 (2) (2008) 145–169, https://doi. org/10.1177/0963662506066719.

Hall, N., Ashworth, P., and Devine-Wright, P.: Societal ac- ceptance of wind farms. Analysis of four common themes across Australian case studies, Energ. Policy, 58, 200–208, https://doi.org/10.1016/j.enpol.2013.03.009, 2013.

Hoen, B., Firestone, J., Rand, J., Elliot, D., Hübner, G., Pohl, J., Wiser, R., Lantz, E., Haac, T. R., and Kaliski, K.: Attitudes of U. S. Wind Turbine Neighbors: Analysis of a Nationwide Survey, *Energ. Policy*, 134, 1–11, https://doi.org/10.1016/j.enpol.2019.110981, 2019.

N.F. Pidgeon, I. Lorenzoni, W. Poortinga, Climate change or nuclear power—No thanks!: A quantitative study of public perceptions and risk framing in Britain, *Global Environ. Change* 18 (1) (2008) 69–85, https://doi.org/10.1016/j. gloenvcha.2007.09.005.

Renn, O.: The Challenge of Integrating Deliberation and Ex- pertise: Participation and Dis- course in Risk Management, in: *Risk Analysis and Society: An Interdisciplinary Characteriza- tion of the Field*, edited by: McDaniels, T. L. and Small, M. J., Cambridge University Press, Cambridge, 289–366, https://doi.org/10.1017/CBO9780511814662, 2004.

Ruddat, M. (2022): Public acceptance of wind energy – concepts, empirical drivers and some open questions. *Wind Energy Science*, 7, S. 1679–1691, 2022 DOI: https://doi.org/10.5194/wes-7-1679-2022.

Sonnberger, M. and Ruddat, M.: Local and socio-political accep- tance of wind farms in Ger- many, Technol. Soc., 51, 56–65, https://doi.org/10.1016/j.techsoc.2017.07.005, 2017.

Sonnberger, M. / Ruddat, M. / Arnold, A. / Scheer, D. / Poortinga, W. / Böhm, G. / Bertoldo, R. / Mays, C. / Pidgeon, N. / Poumadère, M. / Steentjes, K. / Tvinnereim, E. (2021): Climate concerned but anti-nuclear: Exploring (dis)approval of nuclear energy in four European coun- tries. *Energy Research & Social Science* 75 (5), pp. 1-12, DOI: https://doi.org/10.1016/j.erss.2021.102008.

A. Spence, D. Venables, N.F. Pidgeon, W. Poortinga, C. Demski, *Public Perceptions of Climate Change and Energy Futures in Britain: Summary Findings of a Survey Conducted in January-March 2010*, Technical Report, Cardiff, 2010.